# A Method to Extract Measurable Indicators of Coastal Cliff Erosion from Topographical Cliff and Beach Profiles: Application to North Norfolk and Suffolk, East England, UK

**Pablo Muñoz López [1], Andrés Payo [1,*], Michael A. Ellis [1], Francisco Criado-Aldeanueva [2] and Gareth Owen Jenkins [1]**

[1] British Geological Survey, Keyworth, Nottingham NG12 5GG, UK; bgsvispmunoz@bgs.ac.uk (P.M.L.); mich3@bgs.ac.uk (M.A.E.); gjenkins@bgs.ac.uk (G.O.J.)

[2] Physical Oceanography Group, Department of Applied Physics II, University of Málaga, 29071 Málaga, Spain; fcaldeanueva@ctima.uma.es

[*] Correspondence: agarcia@bgs.ac.uk; Tel.: +44-0115-936-3103

**Abstract:** Recession of coastal cliffs (bluffs) is a significant problem globally, as around 80% of Earth's coastlines are classified as sea cliffs. It has long been recognised that beaches control wave energy dissipation on the foreshore and, as a result, can provide protection from shoreline and cliff erosion. However, there have been few studies that have quantified the relationship between beach levels and cliff recession rates. One of the few quantitative studies has shown that there is a measurable relationship between the beach thickness (or beach wedge area (BWA) as a proxy for beach thickness) and the annual cliff top recession rate along the undefended coast of North Norfolk and Suffolk in eastern England, United Kingdom (UK). Additionally, previous studies also found that for profiles with low BWA, the annual cliff top recession rate frequency distribution follows a bimodal distribution. This observation suggests that as BWA increases, not only does cliff top recession rate become lower, but also more predictable, which has important implications for coastal stakeholders particularly for planning purposes at decadal and longer time scales. In this study, we have addressed some of the limitations of the previous analysis to make it more transferable to other study sites and applicable to longer time scales. In particular, we have automatised the extraction of cliff tops, toe locations, and BWA from elevation profiles. Most importantly, we have verified the basic assumption of space-for-time substitution in three different ways: (1) Extending the number or years analysed in a previous study from 11 to 24 years, (2) extending the number of locations at which cliff top recession rate and BWA are calculated, and (3) exploring the assumption of surface material remaining unchanged over time by using innovative 3D subsurface modelling. The present study contributes to our understanding of a poorly known aspect of cliff–beach interaction and outlines a quantitative approach that allows for simple analysis of widely available topographical elevation profiles, enabling the extraction of measurable indicators of coastal erosion.

**Keywords:** cliff behaviour; cliff recession; erosion; beach wedge area (BWA); space-for-time substitution; ergodic system

## 1. Introduction

Recession of coastal cliffs (bluffs) is a significant problem globally, as around 80% of Earth's coastlines are classified as sea cliffs [1]. Coastal cliffs are formed as erosional landforms by weathering and erosion where the cliff, beach, and shore platform dynamically interact, affecting the rate of change

driven by climatic forcing [2]. It has long been recognised that beaches control wave energy dissipation on the foreshore and, as a result, can provide protection from shoreline erosion. However, there have been few studies that have quantified the relationship between beach levels and cliff recession rates. Following a data driven analysis, Lee [3] has shown that there is a measurable relationship between the beach thickness and the annual cliff top recession rate.

Lee used the Beach Wedge Area (BWA) as a proxy for beach thickness and applied it to 18 unprotected soft rock cliff locations in North Norfolk and Suffolk, eastern England, United Kingdom (UK). The BWA was calculated from beach elevation profile data as the beach area above the Mean High Water Spring level (MHWS). Lee showed that for an 11 year period—from 1992 to 2003—there was a non-linear increase in the annual cliff top recession rate as the BWA decreased. In addition, Lee also found that for profiles with low BWA (i.e., less than 5 m$^2$ for North Norfolk profiles and less than 15 m$^2$ for Suffolk profiles), the annual cliff top recession rate frequency distribution follows a bimodal distribution (Figures 10 and 11 in Lee, 2008). This bimodality indicates that, for any given year, the cliff top recession rate could range from less than 1 m/year to as high as 20 m/year. As BWA increases, this bimodality in the cliff top recession rate frequency distribution disappears to become unimodal, with mean values decreasing as BWA increases. This observation suggests that as BWA increases, not only does cliff top recession rate become smaller, but also more predictable, which has important implications for coastal stakeholders—particularly for planning purposes at decadal and longer time scales. For example, the cliff top recession rate predictability of a cliff section with initially low BWA might increase over time as BWA increases and the initial bimodal frequency distribution becomes more unimodal. Conversely, a cliff section with initially large BWA and historical unimodal cliff recession rates could become bimodal if the BWA decreases over time and therefore becomes more unpredictable.

Lee's analysis has some limitations which constrain the transferability of his observation to longer time scales and to other locations. Critically, Lee assumed—without proving it—the ergodic hypothesis (i.e., substituting space for time, so that the dataset represents the response of a single hypothetical site to changes in BWA). Using this assumption, Lee is then able to translate the annual cliff recession rates among 18 sites for a period of 11 years into an arguably temporal sequence of explanatory and predictive value for a hypothetical composite site. A stochastic process is said to be ergodic if its statistical properties can be deduced from a single, sufficiently long, random sample of the process. It should be noted that ergodic systems need to obey the stationary stochastic process, which translates into the basic assumption of space-for-time substitution that the geomorphic evolution of the landform object must have a long-term one-way trend, i.e., the function:

$$G = F(A, M, T),\qquad(1)$$

which is approximately monotone [4]. Equation (1) is a function expression of space-for-time substitution, which states that the landform $G$ (i.e., a coastal cliff) can be regarded as a function, $F$, of the driving force $A$ (i.e., wave energy reaching the surf zone), surface material $M$ (i.e., topography and a mixture of fine sand and gravel in a consolidated or non-consolidated state), and time $T$. Any change in the independent variables $(A, M, T)$ can result in a change in the landform shape. At the same time, the combination of different independent variables will correspond to different landform types. The basic principle of space-for-time substitution theory can be deduced from the inversion of the function $F$ [4]. Assuming that the driving force, original topography, and surface material composition are unchanged (or approximately unchanged), with the passage of time, the topography of a coastal cliff reflects the evolution time, i.e.,

$$T = F^{-1}(G).\qquad(2)$$

If there is a set of spatial sequence landforms $(G_1, G_2, G_3, \ldots, G_n)$ under similar conditions of geological force $A$ and surface material $M$, that landform spatial sequence can be regarded as the evolutionary sequence of this type of landform changing with time.

The main aim of this study is to address some of the limitations of Lee's analysis by transferring his methodology and observations to longer time scales and to other locations. To achieve this we have: (1) Tested the basic assumption of space-for-time substitution and (2) tested repeatability of the method and sensitivity of the results to changes in MHWS.

We start by presenting the study sites of North Norfolk and Suffolk along the east coast of England, UK. As it was not clear from [3] how the cliff top and BWA were calculated, we described how the cliff top, toe, and BWA areas are extracted from elevation profile data, and provided the scripts used as Supplementary Materials for anyone interested in repeating the analysis elsewhere. We also assessed the sensitivity of the BWA and cliff top recession relationship to changes in the values of the MHWS used as the lower bound of the BWA. We used the new MHWS provided by the Environment Agency Coastal Flood Boundary Dataset update (CFB Update 2018, [5]). To test the basic assumption of space-for-time substitution, we extended the time span of the analysis from 11 to 24 years (i.e., 1993 to 2018) for the same 18 sites, and verified that the relationship between cliff top recession rate and BWA remains valid. We were also able to test the robustness of the analysis by increasing the number of study site transects using several Digital Terrain Models (DTM) of the study area over different years. We also verified the implicit assumption of surface material remaining unchanged over time. We did this by combining innovative 3D modelling of the superficial deposits and a non-intrusive survey method. We conclude the analysis by noting implications of the results obtained for coastal stakeholders in the study area and elsewhere.

## 2. Material and Methods

### 2.1. Study Site

The East Anglian coast (Figure 1a) is composed of Pleistocene clays and poorly consolidated sand and gravel, which form a series of cliffs separated by lowlands, marshes, and small estuaries [3]. Most of the coastline along the study area in North Norfolk and Suffolk is defended by hard defences, but our study focuses on the undefended cliff sections. The area is exposed to the waves generated in the North Sea, with dominant wave directions coming from the north for North Norfolk and the south and northeast for Suffolk (Figure 1d,e) [6]. The hundred-year return period wave height in the study locations ranges from 4 to 5 m in North Norfolk and ca. 3 m in Suffolk [7,8].

The North Norfolk cliffs are 15–60 m high and cover 25 km of coastline (Figure 1b). The bedrock geology is composed of Neogene and Quaternary poorly consolidated rocks and sediments, consisting of gravel, sand, silt, and clay. The superficial deposits are mainly composed of glacial sand, gravel, and till [9]. The beaches are mostly composed of sand (∼0.25 mm) and gravel (>2 mm), and shingle to a lesser extent [10]. The transport of sediments occurs both to the west and east [11,12], but drift to the east predominates. The area's defences (seawalls and timber palisades) act as sediment retention obstacles [13], which can influence coastal dynamics by preventing shoreline recession and retaining sediment that otherwise will be lost due to strong gradients on the alongshore current.

The Suffolk cliffs are 5–20 m high and cover 15 km of coastline (Figure 1c). The bedrock geology is similar to that of North Norfolk, and the superficial deposits are similar, with the addition of alluvium deposits composed of clay, silt, and sand. The Crag Group is also more commonly exposed along the coastline [9]. The beaches are predominantly composed of shingle (>2 mm), and less commonly with sand (0.063–2.0 mm) [10]. Sediment transport is mostly to the south, with an exceptional reversal around Benacre Ness [8,11]. Some of the profiles along the Suffolk coastline have not been included in the analysis in order to maintain the ergodic nature of the experiment:

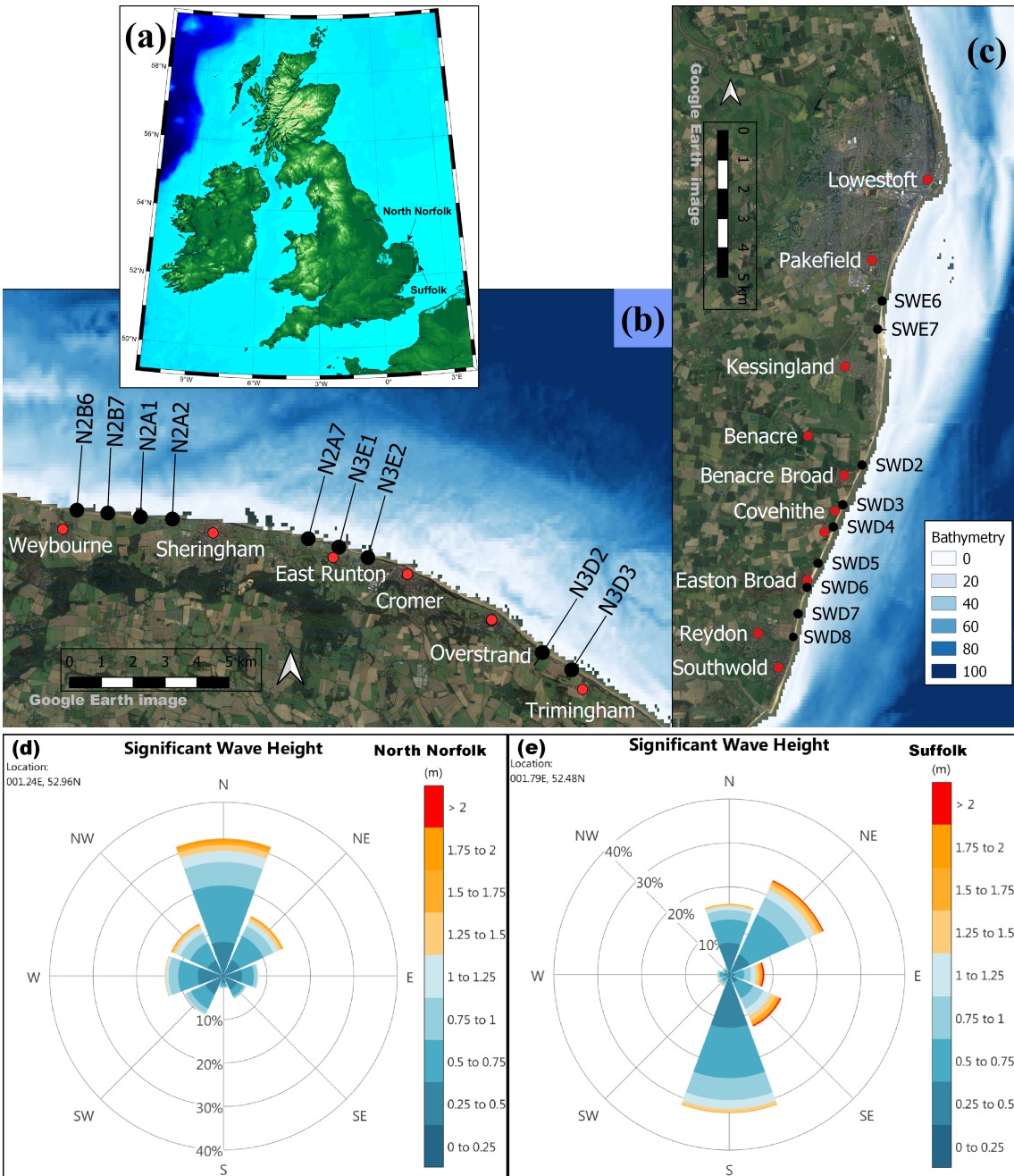

**Figure 1.** (**a**) General location of cliff lines. (**b**) North Norfolk cliffs and profile locations. (**c**) Suffolk cliffs and profile locations. Wave rose for North Norfolk (**d**) and Suffolk (**e**) areas [6].

- Profile SWE6 has a large period of missing data from 2002 to 2014 and is therefore excluded from the long-term analysis.
- Profile SWE7 showed a strong erosion trend from 1991 to summer 2002, followed by a period of increasing accretion. Aerial images show that this is due to the northerly migration of Benacre Ness, building the beach outwards. A reversal of the trend of accretion took place after summer 2002 until 2007, and in 2011, it reverted back to eroding [14,15].
- Profile SWD8 is can be classified as defended, as a new rock groyne was placed in 2006, together with concrete tripod blocks in front of the cliff line and private works, to counteract the outflanking of the sea wall placed in 2003 to 2005 [16]. Therefore, it is excluded from the long-term analysis.

### 2.2. Topographic Elevation Profiles and MHWS Database

We used the bi-annual topographic elevation profiles, available from the Channel Coastal Observatory (CCO, [17]). The beach profiles were measured from a permanent metal marker which is usually set into the sea wall, embankment, promenade, or other hard surface [18]. The required vertical accuracies for survey sections of the profile were ±30 mm on hard surfaces and ±50 mm on soft surfaces. The required horizontal accuracy for the survey was ±0.2 m for sections of the profile surveyed by land. Checks on the consistency of the land survey were undertaken for every profile. Environment Agency (EA) editing of beach data is restricted to removing erroneous spikes, deletion of unwanted points along the profile, and removal of isolated spurious position lines. We downloaded the topographic profiles corresponding to North Norfolk and Suffolk coasts (Figure 1) for all available years (1993–2018). The downloaded profiles were then screened to avoid duplicates. To ensure that we used the same locations as Lee [3], we used the unchanged unique identifier contained in the CCO downloaded files. The old names reported by Lee and the unique identifier are shown in Table A1. We found that some years of the data downloaded were missing or incomplete (see Figure A1).

We used the recently updated MHWS data from the UK Coastal Flood Forecasting (EA—2018 [5]). The MHWS database covers all of the open coastline around the UK and some islands, with a spatial resolution of about 2 km along the open coast or smaller in estuaries and harbours [7]. For each profile, we selected the MHWS value of the nearest point (Table A1). These values are slightly different from the ones used by [3] and directly affect the BWA values. It was not clear from [3] which MHWS values have been used for each profile, as Lee only indicated that in North Norfolk, the MHWS decreases eastward from 2.55 m at Weyborne to 2.45 m at Cromer and 2.14 m at Mundesley, and in Suffolk, increases southward from 0.9 m at Lowestoft to 1.1 m at Southwold. For the validation of our method to extract BWA and cliff top recession rates, we took the mean value between Weyborne to Cromer for North Norfolk (2.45 m) and for the two locations of Suffolk (1 m) as the MHWS to validate our methodology.

### 2.3. Cliff Topcliff top Recession and BWA Extraction from Profile Elevation and MHWS Datasets

In this section, we describe how we extracted the cliff top, cliff toe, and annual BWA (Figure 2a) from the topographic elevation profiles and MHWS databases. To validate the methodology, we compared our recession rate and BWA results with the ones obtained by [3], using the same profiles and historical data time intervals (1992–2003). Once the method was validated, we performed the analysis for the whole available period (1993–2018) and the updated MHWS.

We extracted the cliff top and toe locations from the topographic elevation profiles following a two-step detrending method, as illustrated in Figure 2. This iterative method is an extension of the method proposed by [19], where the cliff top and toe were extracted as the maximum and minimum, respectively, from the detrended elevation profile. We noticed that for some geometries, the cliff top and toe locations might differ from what intuitively would have been noted by a human (i.e., this was also noticed in [19]), resulting in unrealistically different cliff top erosion rates from the ones reported by Lee [3]. This misplacement of the cliff top and toe is due to the sensitivity of the detrended maximum and minimum horizontal locations to the landward and seaward limits used for detrending. In particular, we noticed that the horizontal location (chainage) of the profile seaward limit for a given transect is not always the same (i.e., it is affected by the location of the water level at the time of profile measurement). Cliff top and toe locations in North Norfolk, where the cliffs are generally higher than those in Suffolk, seem more sensitive to the seaward limit chosen to detrend the profile. Therefore, for North Norfolk profiles, we chose the maximum chainage that has been consistently measured for all dates as the seaward limit for the first detrending. For Suffolk profiles, we simply used the seaward end of the profile as the seaward point for detrending. For the landward point used for the first detrending for both sites (North Norfolk and Suffolk), we used the most landward measured point. For the second profile detrending, we replaced the landward point for detrending with the cliff top location obtained from the first detrending. We also noticed that some cliffs have steps which might

give an incorrect cliff toe. To avoid this problem, we checked that the value of BWA obtained was less than a BWA limit of 80 m$^2$ previously obtained by [3] and a first analysis. Figure 3 illustrates how the cliff top and toe locations vary for different profile geometries from North Norfolk and Suffolk.

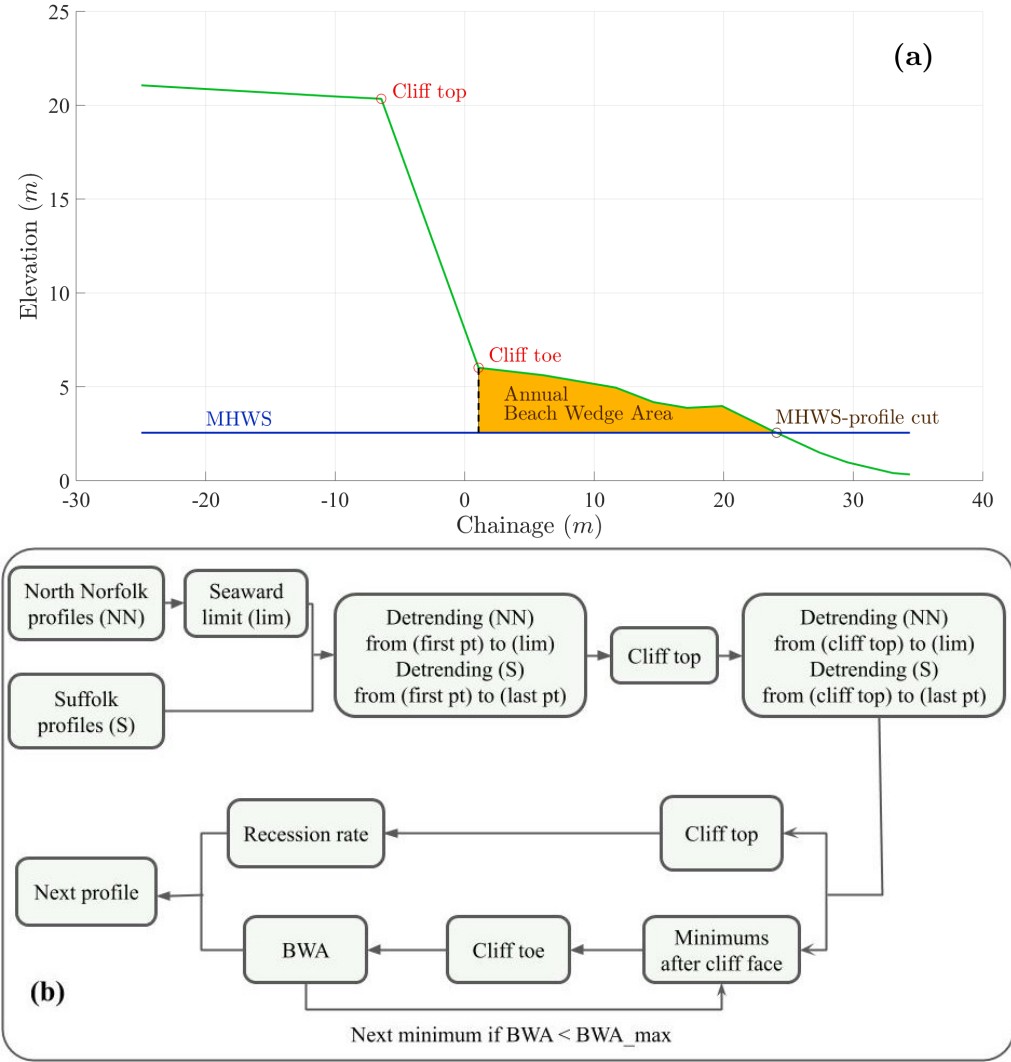

**Figure 2.** (**a**) Cliff top and toe, Mean High Water Spring (MHWS), MHWS profile cut, and annual beach wedge area. (**b**) Flow diagram method.

The recession rate is calculated as the change in the position of the cliff top between successive winter to winter surveys [3], the annual beach wedge area is calculated as the area formed by the intersection of the MHWS line and a vertical line projected down from the cliff toe, and the profile curve is calculated using the trapezoidal numerical integration. The beach wedge area (BWA) is calculated as the mean value of the same two consecutive years used for recession rate.

The calculation of BWA is sensitive to the MHWS value used. According to [3], in North Norfolk, the MHWS decreases eastward from 2.55 m at Weyborne to 2.45 m at Cromer and 2.14 m at Mundesley, and in Suffolk, it increases southward from 0.9 m at Lowestoft to 1.1 m at Southwold. We took the mean value between Weyborne to Cromer for North Norfolk (2.45 m) and for the two locations of Suffolk (1 m) as the MHWS to validate our methodology, and we compared the BWA values with the updated MHWS from 2017.

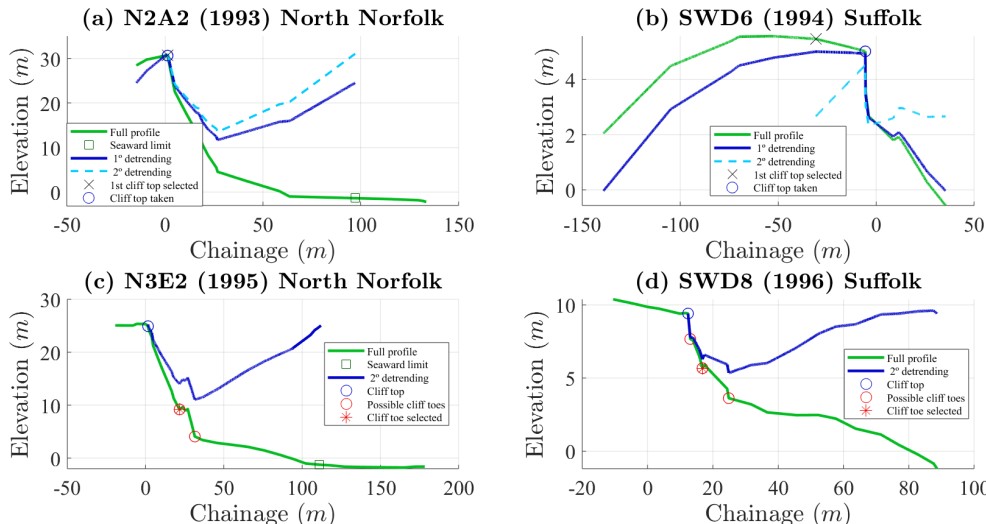

**Figure 3.** Examples of cliff top and toe extraction. (**a**) Cliff top extraction for North Norfolk (profile N2A2, 1993), where the iterative detrending and the seaward limit are shown. (**b**) Cliff top extraction for Suffolk (profile SWD6, 1994), where the iterative detrending is shown. (**c**) Cliff toe extraction for North Norfolk (profile N3E2, 1995), where the iterative cliff toe calculation and the seaward limit are shown. (**d**) Cliff toe extraction for Suffolk (profile SWD8, 1996), where the iterative cliff toe calculation is shown.

### 2.4. Testing the Space-for-Time Substitution Approach Using Evolving Digital Terrain Models

We have used existing Digital Terrain Models (DTM) of the study area to further test the accuracy of the space-for-time substitution. We delineated a new sub-set of transects along the North Norfolk and Suffolk undefended coastlines, extracted the elevation at different consecutive years, and calculated the BWA and annual cliff top recession rate. If the space-for-time approach is valid, the new cliff top recession rate and BWA will follow the general relationship, suggesting that the new subset represents different stages of development of the cliff–beach complex. We used time-stamped-LiDAR DTM produced by the EA at 1 m raster resolution. These raster elevation models cover more than 60% of England at spatial resolutions of between 0.25 to 2 m and are publicly available (EA, LiDAR-DTM database [20]). We extracted the most recent available data from adjoining years covering the study areas: In North Norfolk, the 2015 and 2017 timestamped DTMs were used for the first six profiles from the east, and 2017–2018 data for the rest, while in Suffolk, the data used are from 2015 to 2017. We delineated the new virtual transects at a horizontal spatial distance on the order of 200 m, with care taken to avoid the defended cliff sections and the non-cliffed sections (e.g., The Broads). The elevation along each transect was extracted using the Profile Plugin v4.1.1 for QGIS (v3.8, https://www.qgis.org/). The cliff top, toe, BWA, and cliff top recession rate were calculated as explained earlier using the Matlab scripts available as Supplementary Information.

### 2.5. Testing the Space-for-Time Substitution Approach Using a 3D Geological Model and Non-Intrusive Passive Seismic Survey

As explained in the introduction, the space-for-time substitution approach implicitly assumes that the driving force, original topography, and surface material composition are unchanged (or approximately unchanged) over time. We tested the assumption of topography and surface material composition being unchanged by looking at the subsurface geology and DTM of the study area. We used an existing 3D subsurface model, which partially covers some of the study area in North Norfolk, developed by [21]. The model covers an area of 83 km$^2$, centred on the village of Trimingham on the Norfolk coast and extending offshore, and is developed on 1:50,000 scale geological maps [22,23]. Based on borehole and coastal cross-section evidence, the glacial deposits are further sub-divided

in the model. The 3D geological model is capped by a composite Digital Terrain Model (DTM) that uses different sources for the onshore and offshore parts of the model. An Environment Agency timestamped LiDAR-DTM from 1999 is used for the onshore part of the model and nearshore, while the multibeam bathymetry data from 2011 is used for the offshore area, both of which are available under the Open Government License. The resulting seamless DTM is sub-sampled to a cell size of five metres. The DTM has an elevation range from +70 to −14 m OD (ordnance datum). From this 3D model, we extracted two sections corresponding to profiles N3D2 and N3D3. On top of these two sections, we plotted the recorded elevation transects for years 1999 and 2018 to better assess if the assumption of topography and surface material remaining approximately unchanged is appropriate.

We also used a non-intrusive survey method to assess beach thickness near profile N3D3. The survey method is the passive seismic survey that has been used previously on the region, but never in the study site [24]. We made these independent observations to increase the confidence in the beach thickness values derived from the 3D geological model. Beach thickness is defined here as the thickness of the contemporary beach, made of loose sand and gravel, over the consolidated shore platform; it plays an important role in the complex feedback processes between the cliff–beach–shore platform dynamics [2]. On 5 November 2019, we sampled the beach thickness at eight locations located approximately 450 m to the south of profile N3D3, as shown in Figure 4.

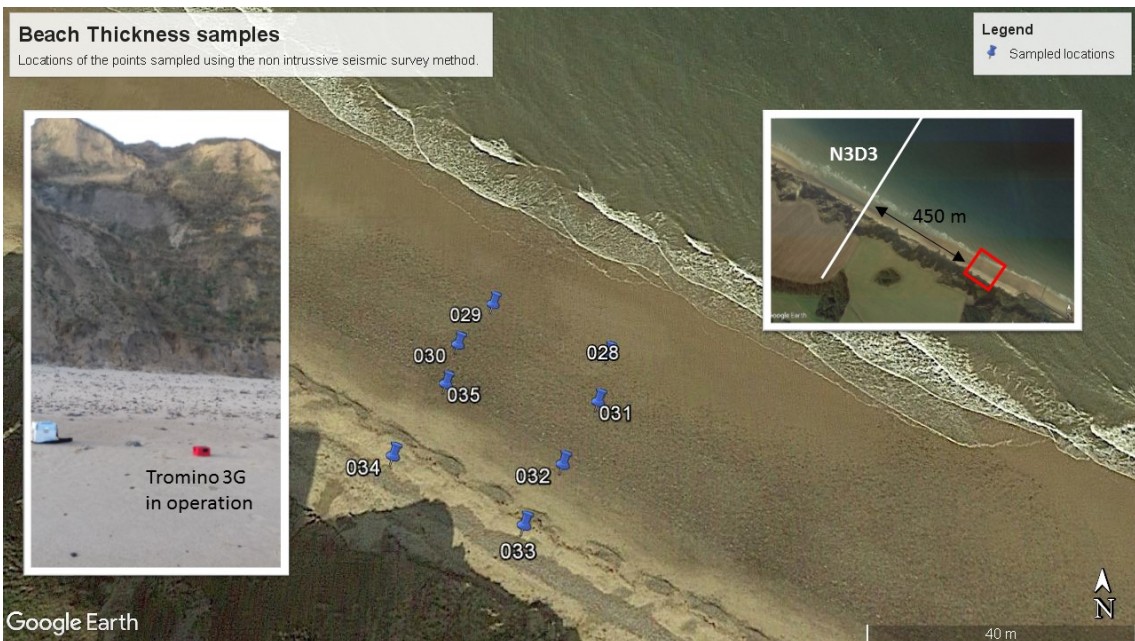

**Figure 4.** Location of the points sampled with the Passive Seismic Survey Method.

Passive seismic surveys measure background seismic noise (both natural <1 Hz and man-made >1 Hz) to estimate the thickness of the different lithologies through different time domains and spectral techniques. Seismic tremor, commonly called seismic "noise", exists everywhere on Earth's surface. It mainly consists of surface waves, which are the elastic waves produced by the constructive interference of P and S waves in the layers near the Earth's surface. Seismic noise is mostly produced by wind and sea waves. Industries and vehicle traffic also locally generate tremor, although essentially at high frequencies (>1 Hz), which are readily attenuated. Passive seismic surveys consist of a series of single-station point recordings, generally arranged into linear transects. These can be of any length and, where organised into an appropriate grid pattern, can be used to generate 3D surfaces of target horizons. Best results are achieved where independent depth control—such as borehole information—is available to calibrate the results. We used the Tromino ENGY-3G (moho.world/tromino), a small (10 × 14 × 8 cm), portable (∼1 kg), broadband, three-component seismometer and the proprietary software Grilla (v7.0), which implements the Horizontal-to-Vertical Spectral Ratio

(H/V) method [25,26]. The reason behind using the spectral noise ratio is that seismic noise varies largely in amplitude as a function of the noise "strength", but the spectral ratio remains essentially unaffected and is tied to local subsoil structure [27]. The Grilla software also provides routines for quality control of the H/V analyses following the European SESAME project directives [28].

Seismic ground noise acts as an excitation function for the specific resonances of the different lithologies in the subsoil. For example, if the subsoil has proper frequencies of 0.8 and 20 Hz, the background seismic noise will excite these frequencies, making them visible when applying the H/V technique on the recordings; these proper frequencies can be used as a proxy for cover thickness. In simple double-layer stratigraphy (superficial cover and bedrock), there is a simple equation [29] relating the resonance fundamental frequency, $f_0$, to the thickness of the layer, $h$, and the shear wave velocity in the same layer, $V_s$:

$$f_0 = \frac{V_s}{4h},$$

(3)

where the value of $V_s$ varies for different materials with typical values of: 100–180 m/s for clay, 180–250 m/s for sand, and 250–500 m/s for gravel. In the case of several peaks on the H/V curve, the peak with the lowest frequency is the fundamental mode ($f_0$, generally the bedrock-cover limit), and other peaks (i.e., $f_1, f_2, \ldots, f_n$) correspond to other geological limits which also cause seismic motion amplification. For the stations located on the beach at the study site, we would expect to see one peak corresponding with the interfaces between the Crag Group and the Chalk ($f_0$), and another peak at the interface between the contemporaneous beach deposits and the Crag Group ($f_1$). The estimated frequencies for $f_0$ and $f_1$ are based on the borehole observations at the study site, indicating that the Chalk surface is at a depth of ca. −20 m OD and maximum Vs velocities are on the order of typical sand deposits and gravel deposits. Based on the expected H/V frequency peaks, the Tromino was set up to measure background noise at a 1024 Hz sampling frequency. According to the Nyquist theorem, the highest frequency that can be recovered from a digitised signal is always lower than half the sampling frequency. Hence, when sampling at 1024 Hz, one can only resolve signals at frequencies up to 512 Hz. For a sand and gravel beach deposit (max Vs ∼500 m/s) of thickness O (0.5 to 1 m), we will expect the $f_1$ peak to be around 125–250 Hz, which is well within the maximum observable frequency when sampling at 1024 Hz. In practice, spectral estimates are statistical in nature, and to have stable results, the observation time should be long enough to comprise at least 10 repetitions of the longest period of interest. For our study, the longest period ($T_0$) (i.e., lower frequencies) corresponds with $f_0$ ∼3 Hz ($T_0 = 1/3 = 0.33$), which means 10 × 0.33 = 3.3 s. Because we have extracted information from seismic noise, we expect fluctuations with time, which can be appropriately controlled by sampling a number N of 3.3 s windows sufficient to compute an average that is statistically significant. Common practice shows this number N to be 30–50, which means in the above example a total recording time of 50 × 3.3 = 165 s = 2.75 min. A total length of 8 min was then considered long enough for this study.

## 3. Results

### 3.1. Repeatability of Lee's Results and Sensitivity to MHWS

Our proposed methodology produces similar results of cliff top recession and BWA values to those reported by Lee (Figure 5). Most (87%) of our estimated values are well-aligned along the perfect agreement line for both study sites (solid line in Figure 5a,b) and fall within the ±2 $\sigma$ confidence bands. Out of 173 cliff top recession rate and BWA values, only 22 values (12 BWA values and 10 cliff top recession rate values) fall outside the ±2 $\sigma$ confidence bands. We checked the outlier values and found different reasons that explain why these values deviate significantly more than the rest. The cliff top recession rates deviate from Lee's values due to some errors in Lee's selection of cliff top (e.g., profile SWD3, 1997 Figure A2), and on several occasions, because we assumed zero erosion rate (i.e., no change) for the years 1992–1993 because of the lack of an elevation profile for 1992 in the EA profile

dataset. The deviation of BWA values seems to be related to our automatic delineation method's not selecting the exact same toe location as Lee (i.e., difficult to prove, as Lee did not provide the cliff toe locations, Figure A3). Small differences between BWA values are related to different methods to determine beach area (i.e., we use the trapezoidal integration under the profile elevation, while Lee used the area of the triangle). BWA values are sensitive to changes in MHWS values used to delimit the beach, but the methods remain similar when using the updated values of MHWS, observing a greater dispersion in the BWA associated with the differences in MHWS (see Table A1 and Figure A4). These results gave us confidence in the repeatability of Lee's results and on the accuracy of our methodology.

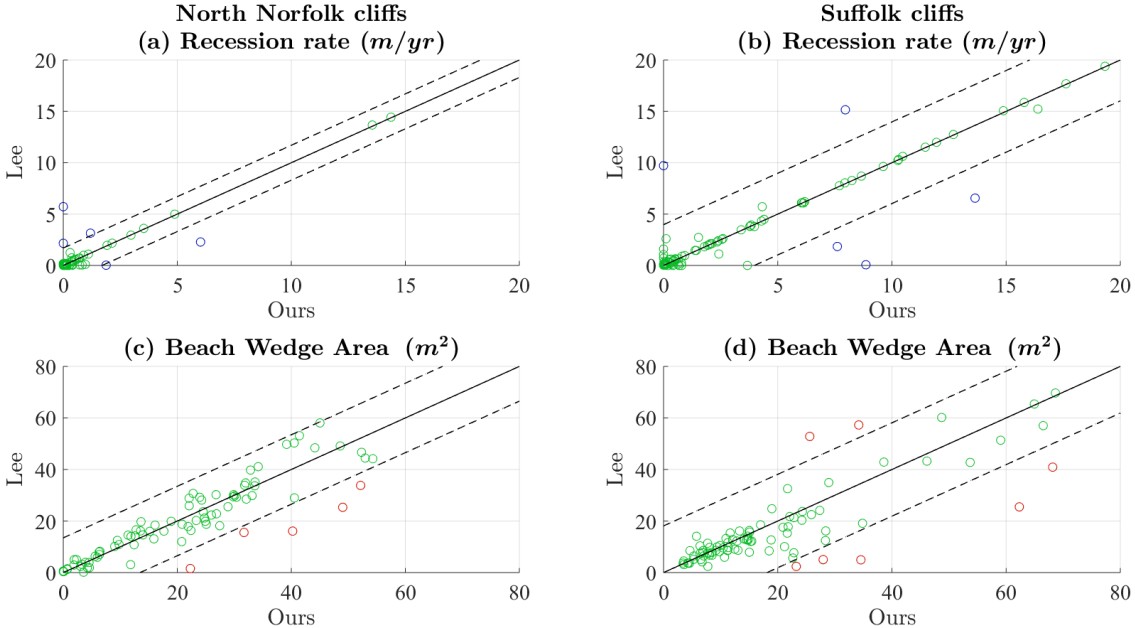

**Figure 5.** Scatter plot using [3] and our calculated cliff top recession rate and beach wedge area (BWA) for North Norfolk (**a**, **c**) and (**b**, **d**) Suffolk cliffs. The blue and red marked points are outliers from recession rate and BWA, respectively. Here, we used a unique MHWS for each location, based on the previous work (see Table A1).

Figure 6 shows the annual scale probability distribution for recession rates, sub-divided into different BWA ranges for [3] and our results for both methods. We have used 1 m/year intervals to bin the cliff top recession data. Using the same BWA thresholds, we observed that the values obtained follow the same trend and practically overlap. We have obtained the same maximum cliff top recession rate value for small and high BWA, while for average BWA values, the maximum cliff top recession rate value obtained by Lee [3] is higher. In particular, for North Norfolk, Lee obtained a recession rate of 13.5 m/year for a BWA of 5.04 $m^2$ (for 1999–2000 at location N3D3), while our maximum value is 6.02 m/year for a BWA of 16.36 $m^2$. For that year and unit, the value obtained for the recession rate is the same, but the calculated BWA differs—which in our case is 4.46 $m^2$, and is therefore classified in the BWA <5 $m^2$ zone. For North Norfolk, the maximum recession rate is 20.6 m/year and for Suffolk is 19.3 m/year, both corresponding to a small BWA. We have shown here that our method reproduces the observed relationships reported by Lee, and we are confident in its applicability to the entire time series.

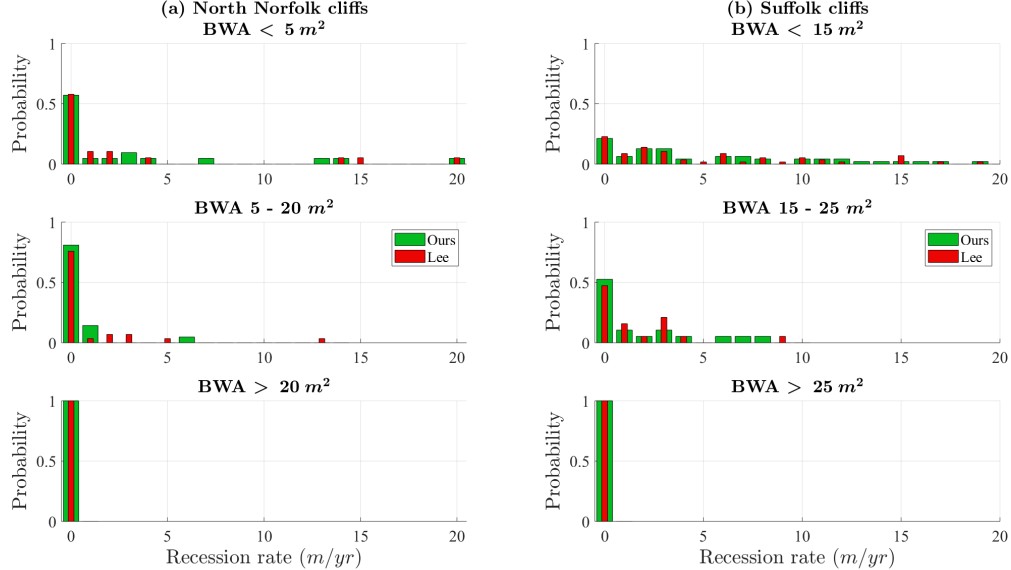

**Figure 6.** Annual scale probability distribution for recession rates sub-divided into different BWA ranges for [3] and our results for North Norfolk (**a**) and Suffolk (**b**) cliffs. The probability histogram is based on intervals of 1 m/year for recession rates. We used the same BWA zone limits of [3] based on the data for a more accurate comparison.

### 3.2. Accuracy of the Space-for-Time Substitution Approach by Extending the Number of Years Analysed

By expanding the study from 11 to 24 years, the three observed behaviours have not changed. Figure 7 shows the annual scale probability distribution for recession rates sub-divided into different BWA ranges using data from 1993 to 2018. The BWA thresholds used vary with those used by [3] because we applied a hierarchical clustering analysis to obtain them (Figure A6), where the results of profile SWE7 since year 2008 have not been taken into account, as this location is a special case that we will discuss later. The three zones are defined again according the new BWA thresholds:

- Zone 1: High recession rates (BWA $<21$ m$^2$)
- Zone 2: Moderate recession rates (BWA 21–52 m$^2$)
- Zone 3: Low recession rates (BWA $>52$ m$^2$)

The new BWA thresholds are comparable to those obtained by [3] and are statistically more reliable: The trend obtained using the old BWA thresholds is the same (Figure A5). Extending the analysis, we observed that the maximum cliff recession rate values increase in all cases except in Zone 2 for North Norfolk due to the change in thresholds (Figure A5), and in Zone 1 for Suffolk and Zone 3 for North Norfolk, where they remain unchanged. The maximum cliff recession rate value is the same for Suffolk (19.33 m/year), while for North Norfolk, it increased (from 20.6 to 22.9 m/year).

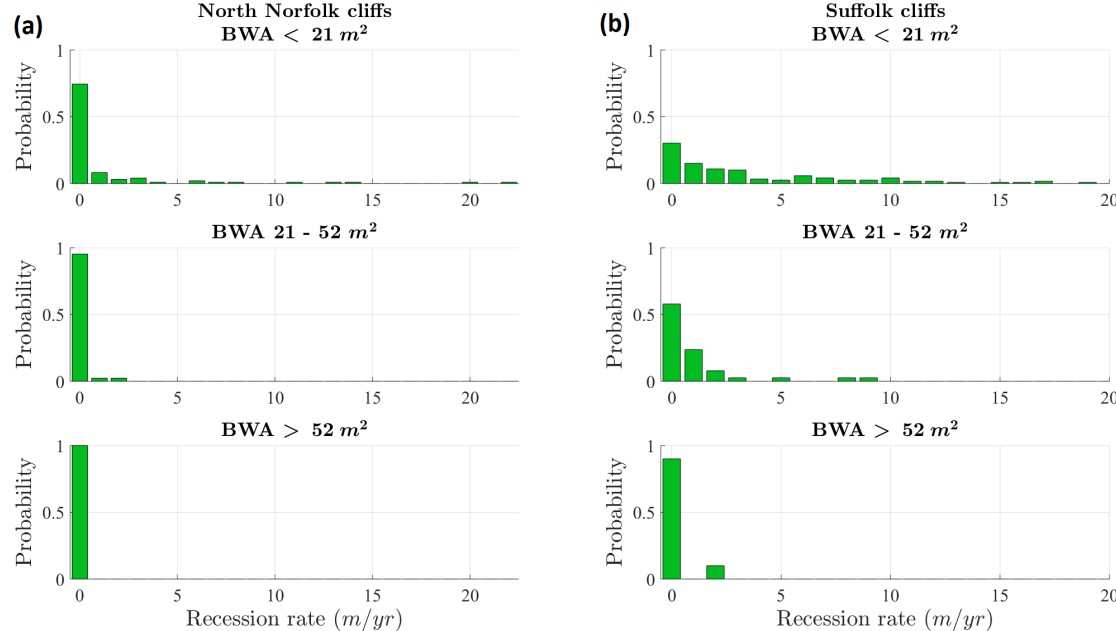

**Figure 7.** Annual scale probability distribution for cliff top recession rates sub-divided into different BWA ranges (based on clustering analysis) for North Norfolk (**a**) and Suffolk (**b**) cliffs using data from 1993 to 2018. The probability histogram is based on intervals of 1 m/year for recession rates. We used the updated MHWS values for the BWA estimates.

Figure 8 shows the annual profile surveys, the BWA, and recession rate calculated for every year available from the profile SWE7 and the Benacre Ness migration. The missing points in Figure 8b are due to incomplete data from years 2010–2011. The profile changed its trend from 2008, increasing the final point of chainage by 150 m (Figure 8a). This trend change is reflected in the BWA, which has increased since 2008, and in the cliff recession that has practically ceased since the same year (Figure 8b). We did not take into account the values obtained since 2008 when making the cluster analysis, because the BWAs were unusually large and the limits obtained did not represent the three zones. However, they are considered when calculating the relationship between BWA and the recession rate, since the trend of Zone 3 (large BWA null–practically null recession rate) is followed.

Figure 9 shows how the cliff top recession rate and BWA values extracted from the new elevation profiles derived from the different DTMs compare with the values obtained from the EA elevation profiles. A total of 54 new elevation profiles (18 for North Norfolk and 36 for Suffolk) were extracted from locations different from the elevation profiles from the EA database. The locations of the new transects and EA profiles are shown in Figure 9a,b. When the values for the new locations are plotted together with the values for the locations of the EA profiles as a scatter plot, it is evident that they follow the same general relationship. For small values of BWA, there is a large range of possible annual cliff top recession rates, ranging from 0 to ca. 7 m/year. As the BWA values increase, the range of possible cliff top recession rates decreases in value. This result suggests that the number of EA profiles (i.e., 17 locations) measured over 24 years was sufficiently large to capture the stochastic relationship between the cliff top recession rate and BWA.

Figure 10 shows the topography and geology for sections N3D2 and N3D3. The EA profile elevation for years 1999 (i.e., same date of the DTM used for the model) and 2018 are overlaid to provide a visual assessment of how the profile topography has changed over time. The cliff is composed of a thick sequence of superficial deposits comprising Anglian glacial sediments, head, and modern coastal and floodplain deposits. Adjacent to these sections, a person standing on the beach will be standing on top of a 2 to 4 m thick layer of contemporaneous marine and beach deposits. Marine and beach deposits are not found at the cliff toe, but in low-lying areas away from the cliff. Underneath

the beach material, there is ca. 10 m of Crag Group which is underlain by more than 20 m of Chalk Group. The topography of the cliff in 2018 looks like a landward translation of the cliff topography in 1999. From these results, it seems that the topography and superficial materials are very similar (i.e., unchanged) for these two transects separated by ca. 1 km.

When the beach thickness was sampled on 5 November 2019, we found that there was a very thin layer of beach material up to the cliff toe (Figure 4). Using the passive seismic survey method, we obtained an average thickness of the contemporaneous beach sediment layer of 0.33 m, and the thickness between the top of the Crag Group and the top of the Chalk Group at ca. 12 m (Table A2), which matched with the values obtained in the 3D model.

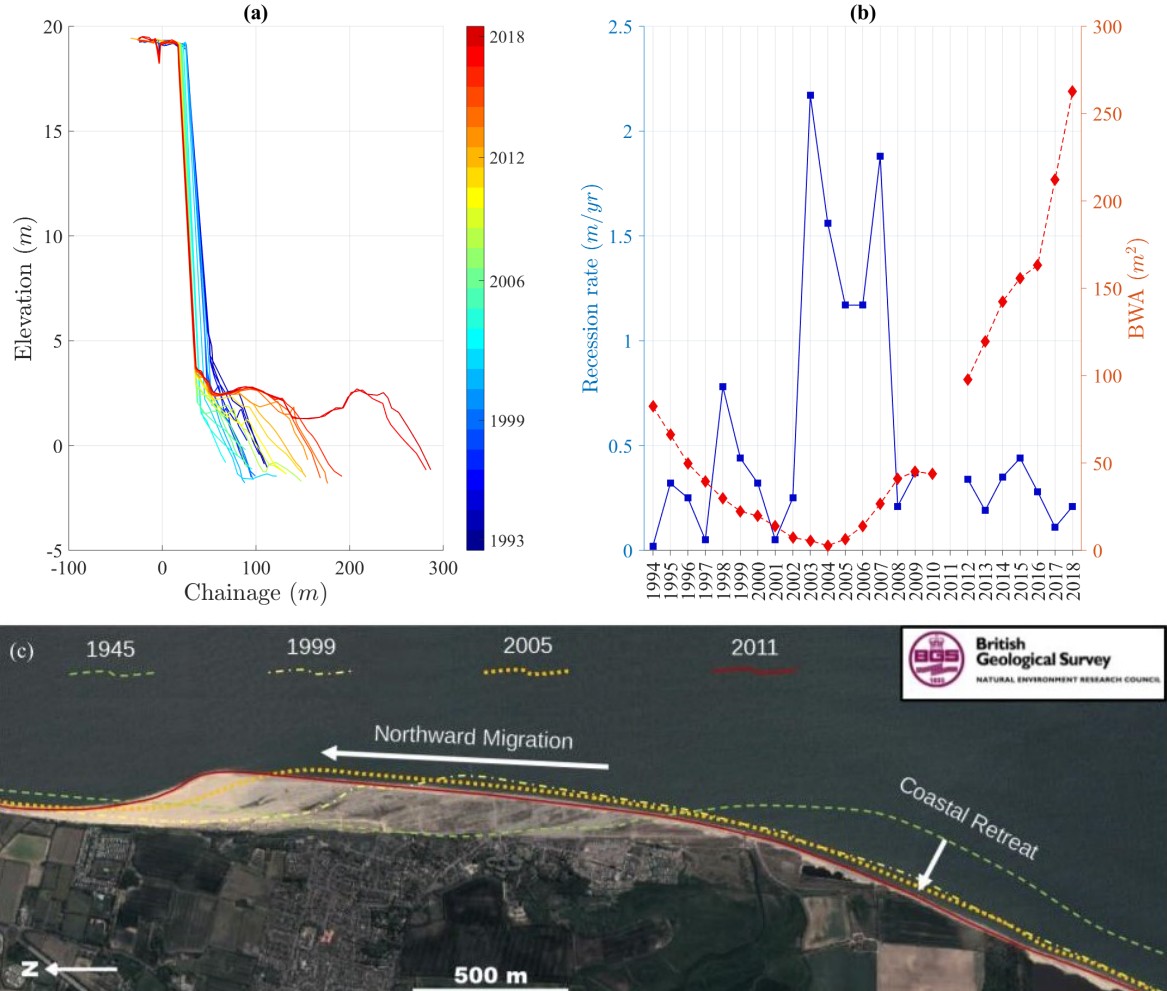

**Figure 8.** (**a**) Annual profile data plot, SWE7 Suffolk. (**b**) BWA (square marked, blue solid line) and cliff top recession rate (diamond marked, dashed red line) calculated for every year for the profile SWE7. The missing points are due to incomplete data from years 2010–2011. (**c**) Benacre Ness northward migration.

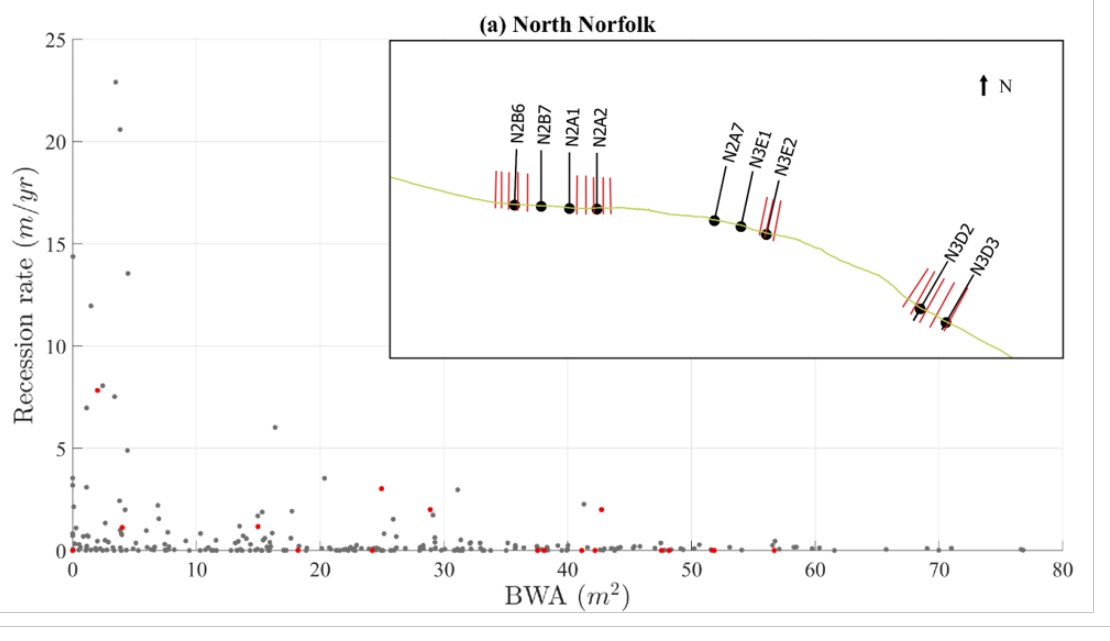

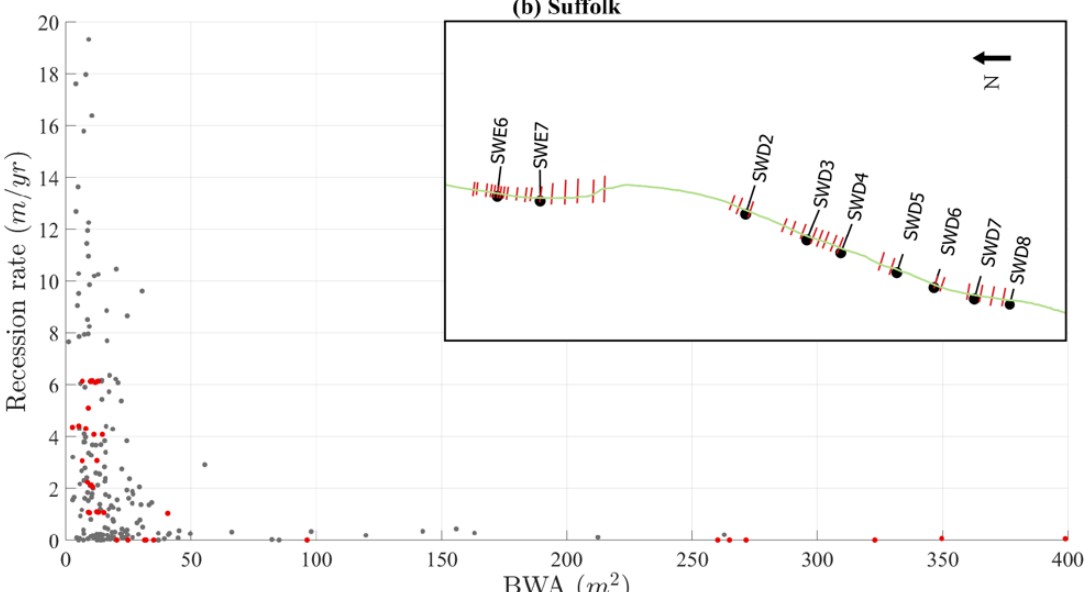

**Figure 9.** Cliff top recession rate and BWA extracted values from the new transects (red dots) compared with those obtained from the Environment Agency (EA) elevation profile database (grey dots). The location of the new transects for Norfolk and Suffolk are shown as red lines on panels (**a**) and (**b**), respectively. Note: The large differences between the maximum BWA areas for the study sites are due to the profiles at the Benacre Ness area.

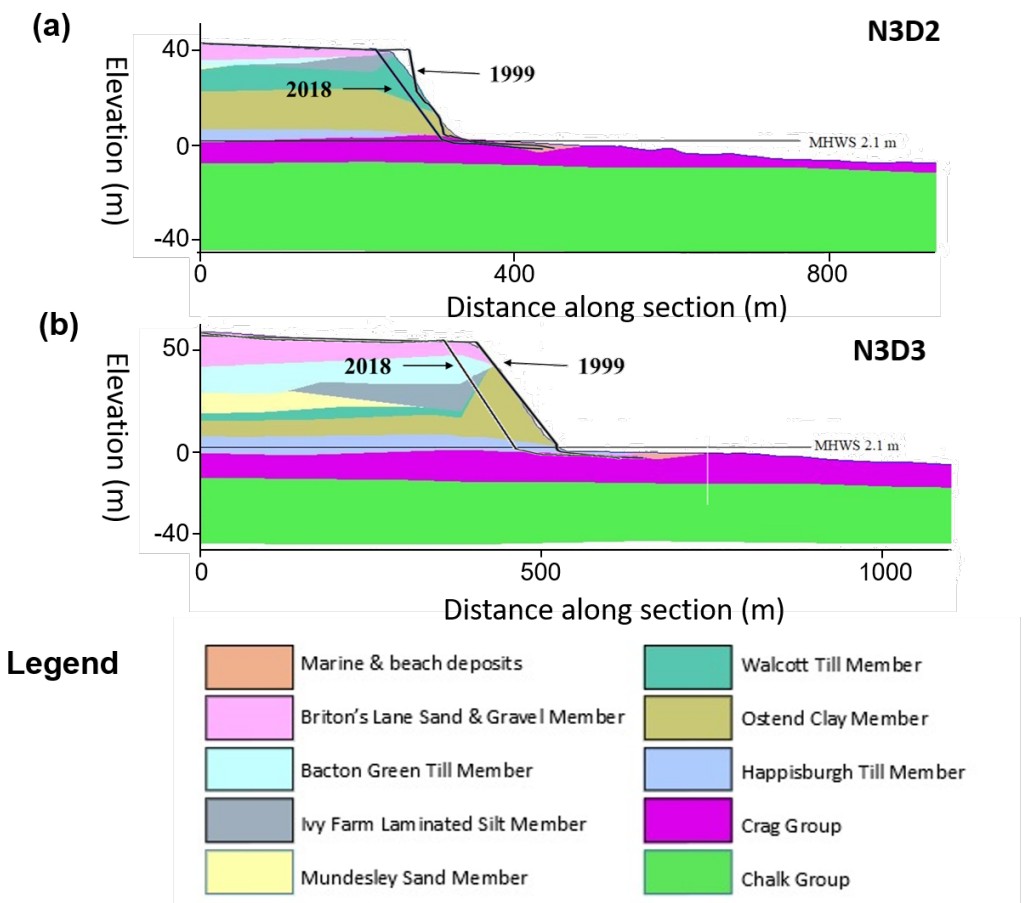

**Figure 10.** Cross-sections extracted from the 3D lithological model at the same locations as transects N3D2 (**a**) and N3D3 (**b**). Topographic profile elevation for years 1999 (same date of the Digital Terrain Model (DTM) used for the model) and 2018 (last date available) overlaid.

## 4. Discussion and Conclusions

It has long been recognised that beaches control wave energy dissipation on the foreshore and, as a result, can provide protection from shoreline erosion. However, there have been few studies that have quantified the relationship between beach levels and cliff recession rates. Following a data-driven analysis, Lee [3] has shown that there is a measurable relationship between the beach thickness (or BWA as a proxy for beach thickness) and the annual cliff top recession rate along the undefended coastlines of North Norfolk and Suffolk in eastern England, UK. Additionally, Lee also found that for profiles with low BWA (i.e., less than 5 m$^2$ for North Norfolk profiles and less than 15 m$^2$ for Suffolk profiles), the annual cliff top recession rate frequency distribution follows a bimodal distribution. This observation suggests that as BWA increases, not only does cliff top recession rate become smaller, but also more predictable, which has important implications for coastal stakeholders, in particular for planning purposes at decadal and longer time scales. For example, the cliff top recession rate predictability of a cliff section with initially low BWA might increase over time as BWA increases and the initial bimodal frequency distribution becomes more unimodal. Conversely, a cliff section with initially large BWA and historical unimodal cliff recession rates could become bimodal if the BWA decreases over time and therefore becomes more unpredictable.

In this work, we have addressed some of Lee's analysis limitations to make it more transferable to other study sites and applicable to longer time scales:

- As it was not clear from [3] how the cliff top and BWA were calculated, we have described how the cliff top, toe, and BWA are extracted from elevation profile data, and provided the scripts

used as Supplementary Materials for anyone interested in repeating the analysis elsewhere. This automatic extraction of the cliff top and toe locations and BWA is an extension of the ClifMetrics approach presented by [19]. In particular, we have shown how cliff top and toe locations can be more accurately estimated by a two-step detrending process of the elevation profiles (Figure 2b). We have demonstrated how this automatic procedure produces similar results to the ones reported by Lee (Figure 5).

- Using this automatic extraction method, we tested the sensitivity of BWA results to changes in MHWS using the EA-updated MHWS values [5], and we found that BWAs are slightly different when using the MHWS used by Lee (Figures 5 and A4).
- For low BWA values, we found that cliff recession rates do not follow a bimodal distribution as suggested by Lee's analysis. We found that for both study sites, the annual cliff recession rate distribution with the most frequent annual erosion rate being less than 1 m/year decays exponentially as BWA increases (Figure 6). We think that the bimodal behaviour reported by Lee was an artefact resulting from plotting the frequency vs. erosion rate data as a continuous distribution instead of discrete bins, as well as slight differences in the calculation of BWA.
- We noticed that the decay of the annual cliff recession rate when BWA increases is more pronounced for North Norfolk profiles (Figure 6a) than for Suffolk profiles (Figure 6b). We speculate that this could be related to the wave angle at breaking being more shore normal at Norfolk and more oblique at Suffolk (Figure 1) but we have not investigated this in any detail in this work.
- We tested the basic assumption of space-for-time substitution which was assumed but not demonstrated by Lee. We have done this by (1) extending the number of years analysed from 11 to 24 (Figure 7), (2) extending the number of locations at which cliff top recession rates and BWAs are calculated (Figure 9), and (3) exploring the assumption of surface material remaining unchanged over time by using innovative 3D subsurface modelling (Figure 10). We found that the assumption that the undefended coastal stretches of North Norfolk and Suffolk are an expression of an ergodic process is supported by the results.
- We used the Passive Seismic Survey method to estimate the beach thickness at Trimingham and found that beach thickness (i.e., depth of the beach loose material on top of the weakly consolidated shore platform) is of the order of 0.3 m, and is clearly not enough to protect the shore platform underneath from eroding during storms where breaking wave heights are ca. 4 m in Norfolk and ca. 3 m in Suffolk.

We acknowledge that our analysis has some limitations regarding the transferability of the proposed method to extract the cliff top, toe, and BWA from elevation profiles (Figure 2). This method is only valid for non-protected cliff lines. In particular, the use of prior knowledge of the maximum expected BWA to refine the location of the cliff toe is required.This constraint was not used for the special case of Benacre Ness (Figure 8). The influence on the results is limited to 21 values (16 at North Norfolk and 5 at Suffolk) out of 394 total values (i.e., less than 5%). We also acknowledge that there are spatial differences in the driving force at the two study sites (i.e., wave energy flux at breaking and tidal range) which have not been considered in this analysis. Finally, we also acknowledge that framing this analysis under the ergodic theory could be contested [4]. While we see the value of providing a theoretical framework, our analysis can also be understood as us seeking a general statistical behaviour from observations.

Our findings have implications for both coastal scientists and stakeholders. The space-for-time substitution approach is normal in coastal geomorphology—as in many other disciplines—but difficult to demonstrate with field data. We have provided three different approaches that could be used elsewhere to test the accuracy of this approach. Coastal stakeholders who manage the risk of cliff erosion along coastal stretches that behave as ergodic systems and have measurements of elevation profiles at different locations over time can now better quantify the threshold BWAs that will provide minimum cliff top retreat and maximum predictability.

**Supplementary Materials:** The following are available online at http://www.mdpi.com/2077-1312/8/1/20/s1, The Matlab scripts used to extract the cliff top, toe and BWA.

**Author Contributions:** Conceptualisation, P.M.L., A.P., M.A.E., and F.C.-A.; Formal analysis, P.M.L.; Funding acquisition, P.M.L. and A.P.; Investigation, P.M.L.; Methodology, P.M.L. and A.P.; Project administration, A.P. and M.A.E.; Resources, A.P. and M.A.E.; Supervision, A.P.; Writing—original draft, P.M.L.; Writing—review and editing, A.P., M.A.E., F.C.-A., and G.O.J. All authors have read and agreed to the published version of the manuscript.

**Funding:** This research was funded by the UK Natural Environment Research Council (NE/M004996/1; BLUE-coast project). P.M.L. would also like to thank the "On the Move" program from the Society of Spanish Researchers in the United Kingdom for funding his stay at the British Geological Survey where this work has been conducted.

**Acknowledgments:** Thank you to all those who assisted with the preparation of this manuscript, in particular to Helen Burke for her support with the 3D geological model, to Andrew Barkwith for the useful discussions during the preparation and analysis of the data, and to Simone Sammartino for the help with the early stage of the Matlab software script. We thank the anonymous reviewers for their careful reading of our manuscript and their comments and suggestions.

**Conflicts of Interest:** The authors declare no conflict of interest.

## Abbreviations

The following abbreviations are used in this manuscript:

| | |
|---|---|
| BWA | Beach Wedge Area |
| MHWS | Mean High Water Spring level |
| EA | Environment Agency |
| CCO | Coastal Channel Observatory |
| BGS | British Geological Survey |
| DTM | Digital Terrain Model |
| OD | Ordnance Datum |

## Appendix A. Tables and Figures

**Table A1.** Location (x,y) in EPSG:28800—OSGB 1936 / British National Grid of the first point of the profile on 1993, unit identifier (Reg_name), old names reported by Lee [3], MHWS of the year 2017 taken from UK Coastal Flood Forecasting (CFB update 2018, [5]), and total recession ($R_t$) of the cliff top from 1993 to 2018. The profile SWD8 was modified anthropogenically in the year 2003, so the total recession is until that year, because after that, the top cliff recession ceased. Note that in Suffolk, the updated values of MHWS have reversed their trend with respect to those used by [3].

| Location | x (m) | y (m) | Reg_name | Old Name | MHWS (m) * | $R_t$ (m) |
|---|---|---|---|---|---|---|
| North Norfolk | 611546 | 343622 | 3b00046 | N2B6 | 2.48 | 9 |
| | 612518 | 343577 | 3b00065 | N2B7 | 2.48 | 6 |
| | 613546 | 343501 | 3b00087 | N2A1 | 2.45 | 5 |
| | 614548 | 343486 | 3b00107 | N2A2 | 2.45 | 2 |
| | 618826 | 343070 | 3b00194 | N2A7 | 2.38 | 24 |
| | 619792 | 342840 | 3b00214 | N3E1 | 2.34 | 4 |
| | 620716 | 342554 | 3b00234 | N3E2 | 2.34 | 6 |
| | 626352 | 339908 | 3b00357 | N3D2 | 2.19 | 81 |
| | 627257 | 339355 | 3b00378 | N3D3 | 2.19 | 56 |
| Suffolk | 653666 | 288957 | 3c00071 | SWE6 | 0.97 | 33 |
| | 653516 | 287990 | 3c00088 | SWE7 | 0.97 | 5 |
| | 653340 | 283329 | 3c00176 | SWD2 | 0.94 | 85 |
| | 652722 | 281959 | 3c00204 | SWD3 | 0.93 | 59 |
| | 652387 | 281210 | 3c00220 | SWD4 | 0.93 | 36 |
| | 651985 | 279932 | 3c00245 | SWD5 | 0.92 | 48 |
| | 651641 | 279095 | 3c00263 | SWD6 | 0.92 | 39 |
| | 651366 | 278192 | 3c00280 | SWD7 | 0.90 | 30 |
| | 651221 | 277401 | 3c00295 | SWD8 | 0.90 | 16 |

* Lee [3] shows that in North Norfolk, the MHWS decreases eastward from 2.55 m at Weyborne to 2.45 m at Cromer and 2.14 m at Mundesley, and in Suffolk, it increased southward from 0.9 m at Lowestoft to 1.1 m at Southwold. We took the mean values between Weyborne and Cromer for North Norfolk (2.45 m) and for the two locations of Suffolk (1 m) as the MHWS to validate the metrics.

**Table A2.** Results from the passive seismic survey method: ID locations shown in Figure 4, lat-lon in WGS 84—WGS84—World Geodetic System 1984, shear wave velocity $V_s$ (considered as the average for typical values in sandy beaches), thickness of Crag Group layer $h_0$, and thickness of beach sediment layer $h_1$ for each location point. The missing value corresponds to an unclear $f_1$ peak in the analysis.

| ID | lat (°) | lon (°) | $V_s$ (m/s) | $h_0$ (m) | $h_1$ (m) |
|---|---|---|---|---|---|
| 028 | 52.9021569807082 | 1.384100988507270 | 225 | 10.8 | 0.28 |
| 029 | 52.9022470023483 | 1.383801000192760 | 225 | 13.1 | 0.41 |
| 030 | 52.9021739959717 | 1.3837160076946 | 225 | 11.3 | 0.30 |
| 031 | 52.9020760115236 | 1.38407299295068 | 225 | 11.7 | 0.28 |
| 032 | 52.9019760154188 | 1.38398598879576 | 225 | 11.6 | 0.37 |
| 033 | 52.9018849879503 | 1.38389697298408 | 225 | 11.4 | 0.32 |
| 034 | 52.9019890073687 | 1.38357703574002 | 225 | 11.9 | - |
| 035 | 52.9021060187370 | 1.38369203545153 | 225 | 12.1 | 0.31 |

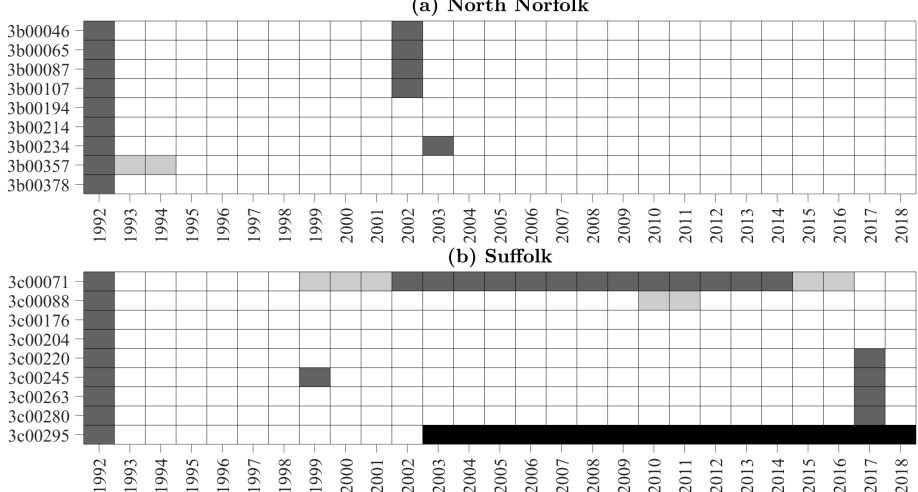

**Figure A1.** Incomplete profiles (light grey), missing profiles (dark grey), and new defence building (black) from the downloaded EA dataset: (**a**) North Norfolk, (**b**) Suffolk.

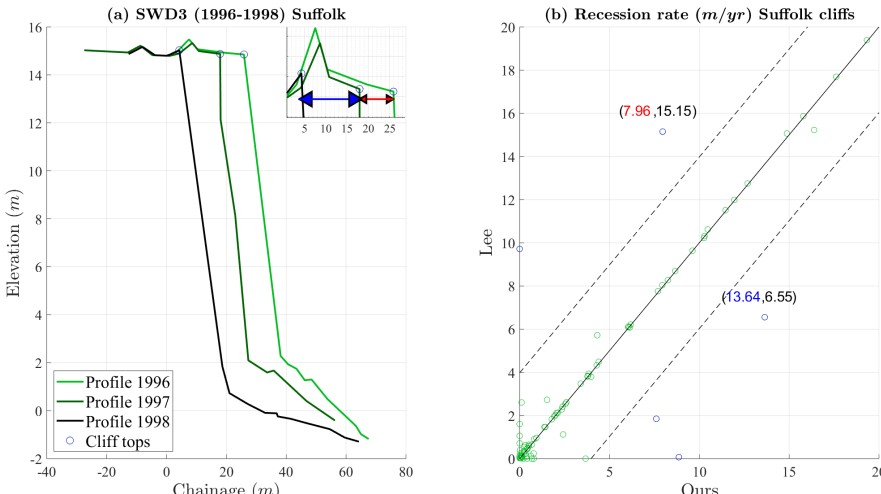

**Figure A2.** Examples of one potential error made in Lee's work [3]. (**a**) Profile plots from SWD3 years 1996–1998 where the cliff tops that we have used are indicated. A zoom in the cliff top area is shown where the top recession can be estimated at first glance. (**b**) Cliff top recession rate scatter plot for Suffolk where the two corresponding outliers with our (red and blue) and Lee's values (black) are shown.

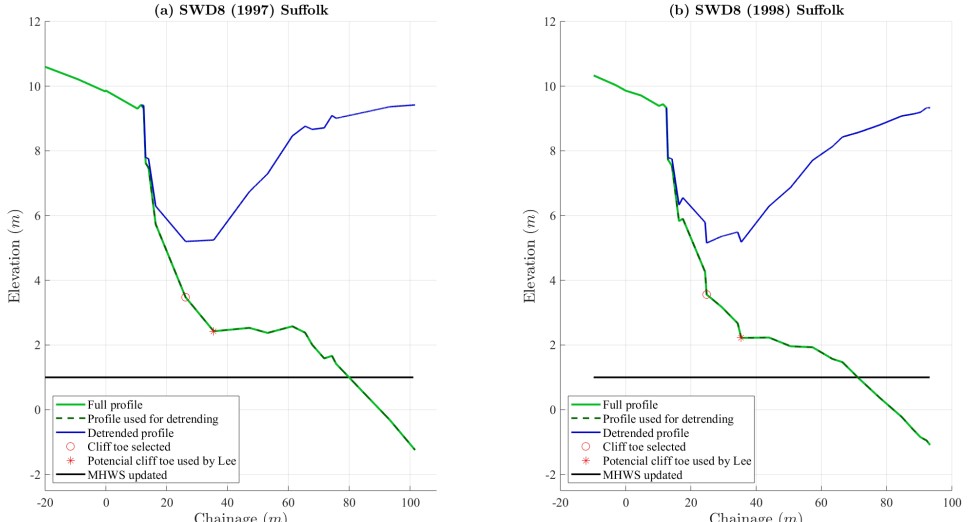

**Figure A3.** Examples of two outliers found in the BWA scatter plot for Suffolk, SWD8. The full profile, the extracted one used for detrending, the detrended profile obtained for extracting the cliff toe, and the updated MHWS are plotted. The cliff toe selected by applying our method is represented by a red circle, and the potential cliff toe used by Lee with an asterisk. The differences between our selected cliff toe and the potential one used by Lee are enough to obtaining significant differences in the BWA.

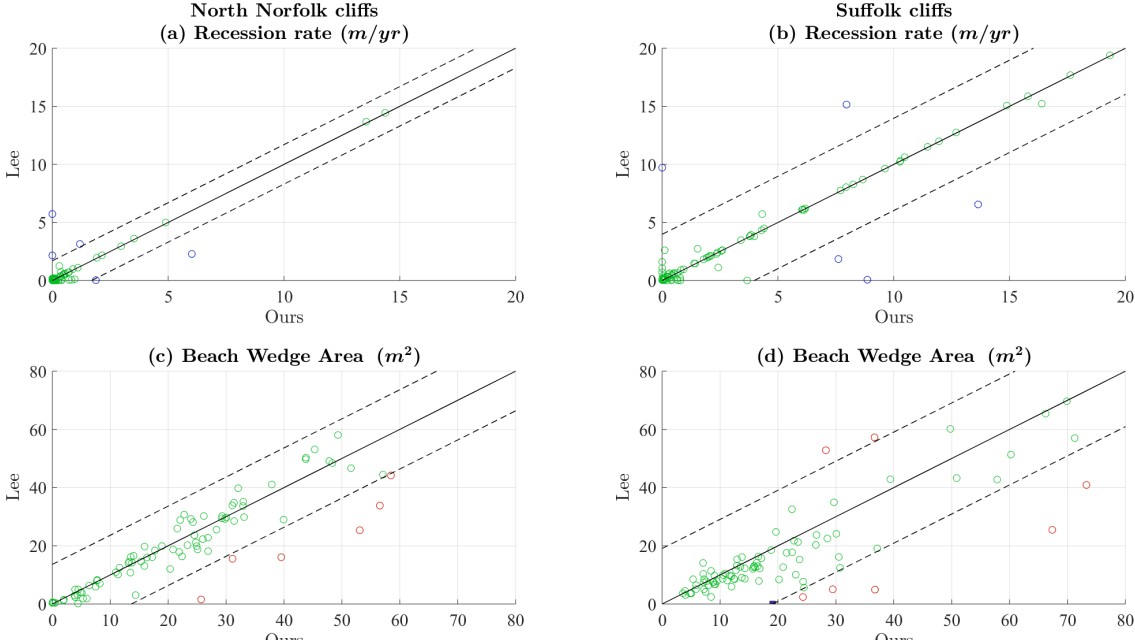

**Figure A4.** Scatter plot using [3] and our calculated cliff top recession rate and BWA for North Norfolk (**a**, **c**) and Suffolk (**b**, **d**) cliffs. The blue and red marked points are outliers from recession rate and BWA, respectively. Here, we used the updated MHWS (see Table A1).

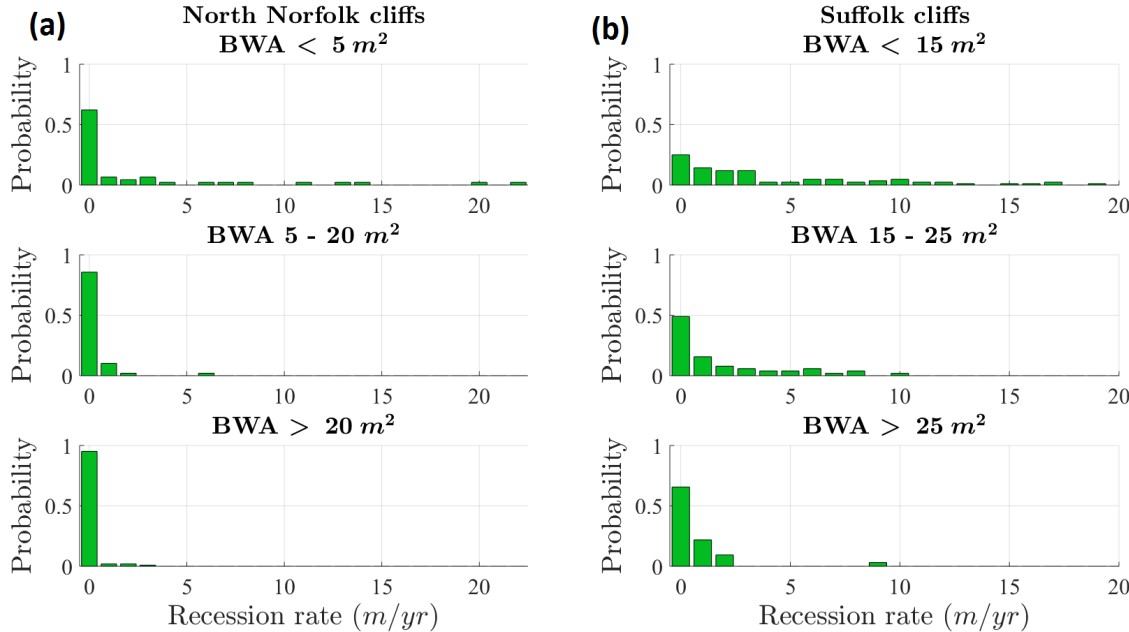

**Figure A5.** Annual Scale probability distribution for cliff top recession rates sub-divided into different BWA ranges based on the data according to [3] for North Norfolk (**a**) and Suffolk (**b**) cliffs. The probability histogram is based on intervals of 1 m/yr for cliff top recession rates.

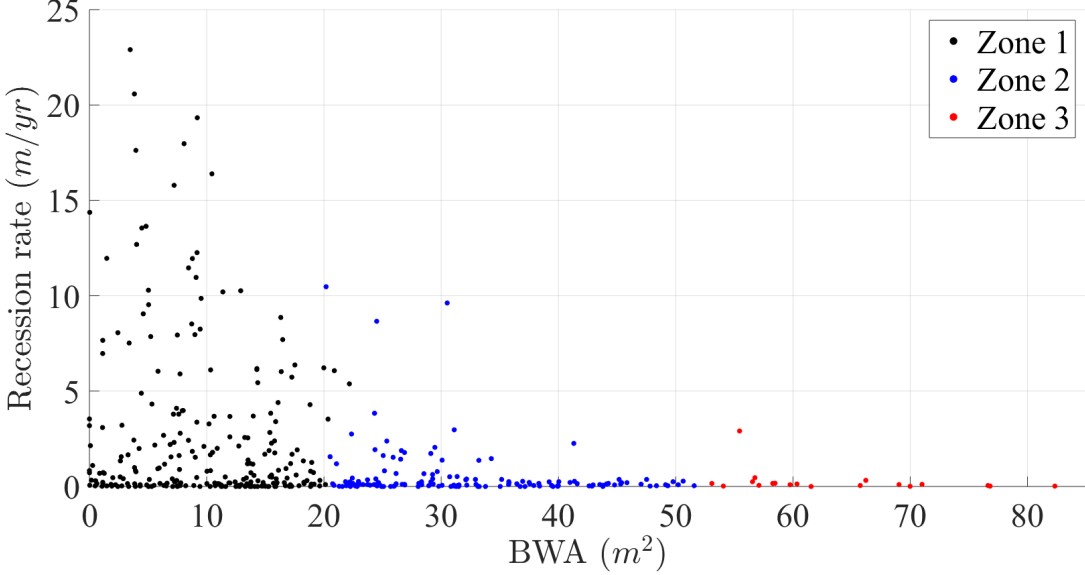

**Figure A6.** Annual scale relationship between BWA and cliff top recession rates for North Norfolk and Suffolk unprotected cliffs from 1993–2018. The colours indicate the three zones obtained using the hierarchical clustering method, where the results of profile SWE7 since the year 2008 have not been taken into account, as this location is a special case related with the Benacre Ness migration.

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
