# Peer review of "A Method to Extract Measurable Indicators of Coastal Cliff Erosion from Topographical Cliff and Beach Profiles: Application to North Norfolk and Suffolk, East England, UK"

_jmse, doi:10.3390/jmse8010020_

Round 1
Reviewer 1 Report
This is an excellent work on cliff erosion modelling and i am suggesting publication on JMSE pending minor corrections.
Figure 3. Please use a the same font size in all legends (a, b, c and d) so they match together.
Author Response
We thank reviewer 1 for his/her positive feedback.
We have amended the captions on Figure 3 as suggested.
Reviewer 2 Report
Title: A method to extract measurable indicators of coastal cliff erosion from topographical cliff and beach profiles: application to North Norfolk and Suffolk,
East England, UK
Authors: Pablo Muñoz López, et al
The manuscript reports the study of the improvement for the analysis methods to study the recession of coastal cliffs.
The paper was well organized and written in English. The analysis study and the sensitivity test are well designed. The improvements make it more transferable to other study sites and applicable to longer time scales.
The referee finds that the manuscript could be accepted for publication as its current form.
Author Response
We thank reviewer 2 for his/her positive feedback and suggestion that the manuscript could be accepted for publication as its current form.
Reviewer 3 Report
This is an interesting paper with some important implications for rapidly eroding, soft rock coasts. It is basically well written, although there are some statements that need to be rewritten (incorrect verb tenses, misuse of capitals, etc). i have attached a file which makes suggestions for some of these changes. The research is sound and the methods are adequately explained.
If I understood you correctly, beach volume data were collected bi-annually (lines 135-136). To what degree do beach volumes on this coast change over a strom-post storm cycle ? In other words, aren't the beach volume data heavily influenced by the period which has elapsed since the last storm, and also the severity of the last storm ?
Lee found that with low beach volumes, rates of cliff recession were bimodal. This was not confirmed in the present study, however, which suggests that recession rates decline asymptopically with increasing beach volume. I would like to see some discussion of possible reasons for both these conclusions as it relates to the possible effects of beach dynamics on wave attenuation, exposure of the cliff base, cliff height and debris production, etc.
As the authors noted, most of this coast is protected, which therefore must limit the relevance of this study to that coast, while applications to other, unprotected coasts is constrained by the effect of local factors in those areas. Some statement regarding the geographical relevance of this study might therefore be appropriate.
